# TOWARDS COMPLEX-QUERY REFERRING IMAGE SEGMENTATION: A NOVEL BENCHMARK

## ABSTRACT

Referring Image Understanding (RIS) has been extensively studied over the past decade, leading to the development of advanced algorithms. However, there has been a lack of research investigating how existing algorithms should be benchmarked with complex language queries, which include more informative descriptions of surrounding objects and backgrounds (*e.g.*, *"the black car."* vs. *"the black car is parking on the road and beside the bus."*). Given the significant improvement in the semantic understanding capability of large pre-trained models, it is crucial to take a step further in RIS by incorporating complex language that resembles real-world applications. To close this gap, building upon the existing RefCOCO and Visual Genome datasets, we propose a new RIS benchmark with complex queries, namely **RIS-CQ**. The RIS-CQ dataset is of high quality and large scale, which challenges the existing RIS with enriched, specific and informative queries, and enables a more realistic scenario of RIS research. Besides, we present a nichetargeting method to better task the RIS-CQ, called dual-modality graph alignment model (**DUMOGA**), which outperforms a series of RIS methods. To provide a valuable foundation for future advancements in the field of RIS with complex queries, we release the datasets, preprocessing and synthetic scripts, and the algorithm implementations at `https://anonymous.4open.science/r/ICLR2024-D652/README.md`.

## 1 INTRODUCTION

Correctly comprehending the subtle alignment (i.e., grounding) between language and vision has long been the pivotal research in the intersecting communities of computer vision and language processing (Yang et al., 2020; 2019; Zhao et al., 2022). Among a range of vision-language grounding topics, Referring Image Segmentation (RIS) has been proposed with the aim to ground a given language query onto a specific region of an image, i.e., typically represented by a segmentation map, as exemplified in Figure 1. By precisely bridging the semantics between the images and texts, RIS plays a crucial role in various downstream cross-modal applications, including image editing (Chen et al., 2018; Li et al., 2020), and language-based human-robot interaction(Wang et al., 2019; Zhang et al., 2020).

Although attracting increasing research attention, the existing RIS benchmarks, unfortunately, can be subject to the **simplicity and naivety of the language queries**. Specifically, via data statistics of the current popular RIS datasets, including RefCOCO and RefCOCO+ (Yu et al., 2016; Mao et al., 2016), we find that *i)* 85.3% text queries have short forms (i.e., with length $\leq 5$ words), and *ii)* 83.8% queries involve only one or two visual objects. We note that such characteristics would inevitably lead to the key pitfalls that hamper the utility and applicability of RIS task.

- On the one hand, caused by the inequality between vision and language, i.e., language is abstract and succinct while vision always entails richer details, simple short language queries with shallow object descriptions usually trigger ambiguity for the visual coreference. The sample in Figure 1 shows the case. More severely, in the existing RIS datasets, a significant majority of query sentences suffer from such ambiguity. Intuitively, RIS models, being trained on such overly simplistic data, would largely fail to disambiguate the image referring and lead to suboptimal performance.

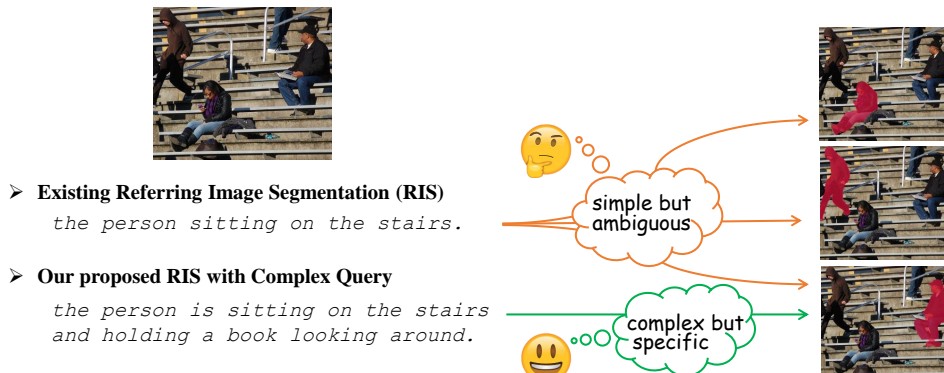

Figure 1: The comparison between the existing Referring Image Segmentation (RIS) and the complex-query RIS proposed in this work.

- On the other hand, in real-world applications, individuals are more likely to input detailed textual descriptions in complex forms, so as to accurately locate the desired objects. Although the recent RIS methods (Yang et al., 2022; Yu et al., 2018; Feng et al., 2021) achieve satisfactory performance on the in-house testing set, they can still struggle when confronted with complex and informative queries. Our preliminary experiment indicates that even the current top-performing RIS model, LAVT(Yang et al., 2022), drops dramatically (74.46 vs. 10.84 in mIoU on two RIS datasets) when testing on the complex query. In other words, being trained on existing RIS datasets can lead to the failure of generalizing to the input queries from realistic users in the wild.

The above observations imperatively motivate the exploration of *Referring Image Segmentation with Complex Queries (RIS-CQ)*. In this work, we propose a novel benchmark for RIS under such complex scenario. Specifically, we construct a RIS-CQ dataset (cf. §2), which is of high quality and large scale. Technically, we first extract the salient objects with their pertaining relations (spatial, action, *etc.*) from an image, and then generate semantically detailed and meaningful descriptions for the referring objects towards their surrounding objects, which serve as the complex-form queries. Notedly, the recent large language models (LLMs), e.g., ChatGPT (Lund & Wang, 2023) have revealed the great capability of human-level understanding of language semantics. Thus we take advantage of the LLM to assist to generate large amounts of complex queries while without sacrificing the labeling quality. Finally, we obtain a novel RIS-CQ dataset with 118,287 images and 13.18 words on average for each query.

The key to accurate RIS recognition in our scenario essentially lies in the deep understanding of the underlying semantics of different modalities, because of the intrinsic complex form of textual query and the sophisticated visual content. To this end, we propose a novel **du**al-**mo**dality **g**raph **a**lignment (dubbed **DuMoGa**) model (cf. §3) to benchmark the RIS-CQ task, where the input sentence and image are represented with the semantic dependency graph (Peters et al., 2018) and semantic scene graph (Tang et al., 2019), respectively. Meanwhile, the semantics of the two modalities should be sufficiently interacted for deep comprehension of the inputs. Correspondingly, two levels of cross-modal alignment learning are carried out in DuMoGa to capture the intrinsic correspondence between the input text and vision, including the structural alignment between two semantic graphs and the feature alignment between two semantic representations. On the RIS-CQ datasets our DuMoGa achieves 24.4 in mIoU, outperforming the current state-of-the-art (SoTA) RIS method by more than 200%.

To sum up, this work contributes to the following three aspects. **(1)** We construct a novel benchmark dataset, RIS-CQ, which challenges the existing RIS with complex queries, and enables a more realistic scenario of RIS research. **(2)** We present a strong-performing system (DuMoGa) to model the task, which brings a new SoTA results on the RIS-CQ dataset. **(3)** Series of in-depth analyses are shown based on our dataset and the systems, where some important and interesting findings are presented and concluded to shed light on the future exploration of this topic. All our data and resource will be made open later to facilitate the follow-up research.

# 2 Constructing Complex Query for Referring Image Segmentation

## 2.1 Problem Definition

Given an image $\mathcal{I}$ and a set of textual queries (i.e., long-form sentences with semantically complex expressions) $\mathcal{Q} = \{\mathbf{p}_i\}_{i=1}^{M}$, RIS-CQ aims to predict a set of segmentation masks $\mathcal{S} = \{\mathbf{s}_i\}_{i=1}^{M}$, where each $\mathbf{s}_i$ correspond to each query $\mathbf{p}_i$ that localizes it in the image. Note that $M$ is the number of referring expressions for a given image $\mathcal{I}$, which is set as $M \in [1, 10]$.

## 2.2 Dataset Construction

In this section, we elaborate on how to elicit complex and contextualized queries from large language models (*i.e.*, ChatGPT) by leveraging diverse structured semantics in images (*e.g.*, inter-object relations). First, we extract holistic semantic relations for each image and then select more significant objects that involve diverse interactions with other objects as the candidate objects. Second, we utilize specially tailored prompts to guide ChatGPT in generating complex queries for each candidate object based on rich visual context. Finally, we manually filter out or revise problematic queries (*e.g.*, ambiguous references), to ensure the annotation quality.

**Step-1: Relation extraction.** To begin with, we utilize an off-the-shelf scene graph generation model, VC-Tree, to directly extract the objects present in the images along with their relationships. These relationships are stored in the form of triplets, such as <person, cup, holding>, which indicates that the person is holding the cup. Additionally, during this step, we apply a simple filter to exclude objects that have fewer than **two** relationships with other objects. We believe that generating queries to describe these objects would be relatively straightforward, increasing the probability of ambiguous references or incorrect references.

**Step-2: Complex query generation.** In this section, we devise a tailored set of prompts to efficiently leverage the in-context learning capabilities of a large language model. For each object and its corresponding relation triplets, we generate a descriptive text query using the model. For instance, given a set of triplets associated with the target object *person*, <person, cup, holding>, <person, wall, leaning on>, <table, person, next to>, ChatGPT can provide us with the output "the person is next to the table, holding the cup and leaning on the wall."

Specifically, we utilize the gpt-3.5-turbo model as our large language model, with the API interface provided by OpenAI. To ensure stable outputs from the model, we set the decoding temperature to 0. The relevant prompts used are detailed in the appendix for reference. An additional noteworthy aspect is that the presence of the large language model entirely liberates us from the manual labor of annotating images for generating language queries. This significantly reduces our workforce costs while ensuring the quality of queries. Moreover, it facilitates the expansion of our dataset to the magnitude of 100k.

**Step-3: Post-processing.** Finally, we manually filter out or revise queries with ambiguous references to ensure a precise one-to-one correspondence between the queries and objects within the image. After multiple iterations in constructing prompts, we have successfully developed prompts that yield high-quality queries generated by ChatGPT. However, it is important to note that since ChatGPT receives input solely from the relation triplets generated by the scene graph model, it lacks certain contextual information from the image. As a result, there may be instances where the generated queries exhibit ambiguous references to the objects being described. Unfortunately, these queries cannot be filtered out automatically in our automated pipeline, necessitating manual intervention for their removal.

## 2.3 Dataset Statistics

Figure 3 presents the statistical analysis of objects and predicates in the RIS-CQ dataset, which consists of a total of 133 object classes and 56 relation classes. The language queries generated are solely based on the objects depicted in Figure 3 along with their corresponding relations. The objects and relations are categorized into 9 groups and 6 groups respectively, with each group containing

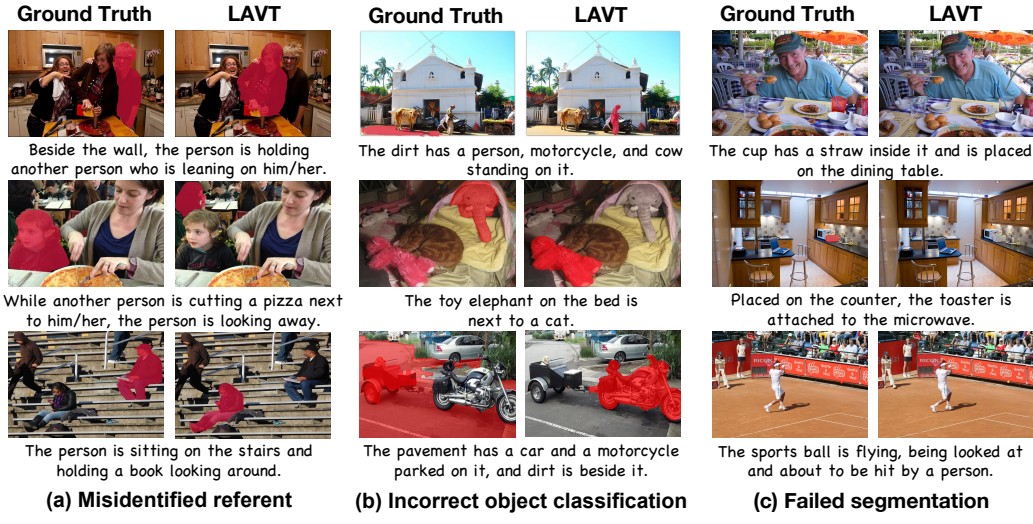

Figure 2: Failure cases of Complex-query Referring Image Segmentation.

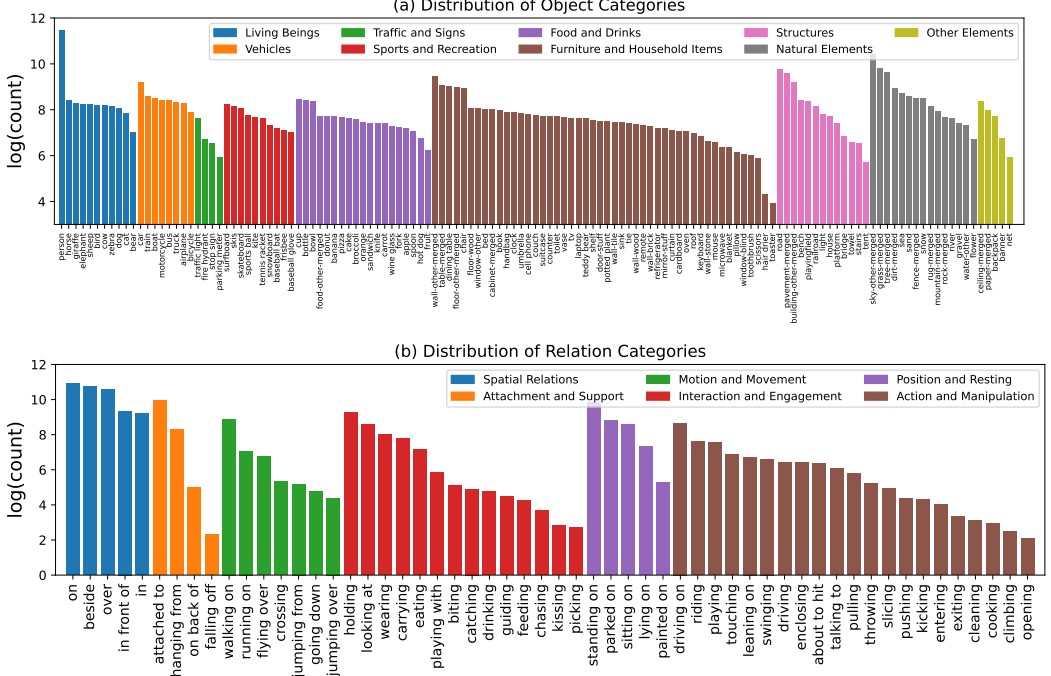

Figure 3: The distribution of object and relation categories, organized based on the parent classes. Best viewed by zooming in.

elements that exhibit certain correlations. The group names represent abstract summaries of the shared attributes among the elements within each group.

We analyze the failure cases of current models on RIS-CQ dataset. From Figure 2, we can summarize them into three types: misidentified referent, incorrect object classification, and failure segmentation.

## 2.4 DATASET COMPARISON

**RefCOCO/ RefCOCO+/ RefCOCOg.** RefCOCO (Yu et al., 2016), RefCOCO+ (Yu et al., 2016), and RefCOCOg (Mao et al., 2016) are three visual grounding datasets with images and referred

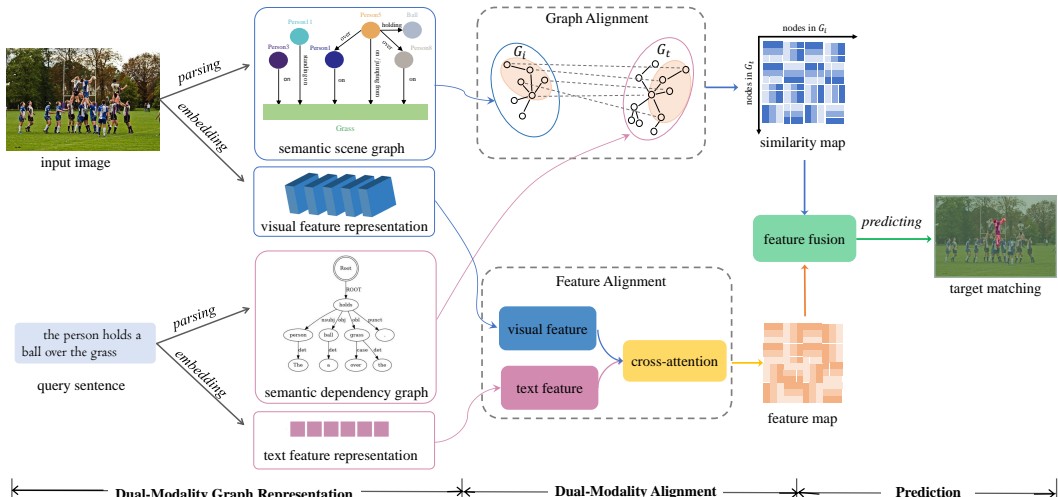

Figure 4: Our proposed DUMOGA framework, includes three procedures: Dual-modality Graph Representation, Dual-modality Graph Alignment, and Prediction. Best viewed by zooming in.

objects selected from MS COCO (Lin et al., 2014). The referred objects are selected from the MS COCO object detection annotations and belong to 80 object classes. RefCOCO (Yu et al., 2016) has 19,994 images with 142,210 referring expressions for 50,000 object instances. RefCOCO+ has 19,992 images with 141,564 referring expressions for 49,856 object instances. RefCOCOg has 25,799 images with 95,010 referring expressions for 49,822 object instances.

**Visual Genome.** Visual Genome (VG v1.4) (Krishna et al., 2017) contains $108,077$ images with $21$ relationships on average per image, which is split into $103,077$ training images and $5,000$ testing images.

Table 1: Statistics of current Referring Image Understanding benchmarks.

| Dataset | RefCOCO | RefCOCO+ | RefCOCOg | Visual Genome | RIS-CQ |
|---|---|---|---|---|---|
| # Images | 19,994 | 19,992 | 26,711 | 108,077 | 118,287 |
| # Text query | 142,209 | 141,564 | 85,474 | 5.4M | 285,781 |
| Avg. query length | 3.61 | 3.53 | 8.43 | 5 | 13.18 |
| Avg. object / query | 1.76 | 1.67 | 3.03 | - | 3.58 |
| Annotation methods | Manual | Manual | Manual | Manual | Auto + Manual |

**RIS-CQ.** is our proposed referring image segmentation benchmark, which targets the explanation of image contents. The image source for RIS-CQ is the union of VG (Krishna et al., 2017) and COCO (Lin et al., 2014) datasets. RIS-CQ challenges RIS models to understand the rich object interactions in daily activities. As shown in Table 1, we summarize the details of each dataset compared with our RIS-CQ dataset.

## 3 PROPOSED METHOD

Unlike classic dense prediction models with complex design in RIS (e.g. LAVT (Yang et al., 2022)), which take expensive computational power to make inferences, the graph learning based architecture provides efficient training process and promising results. To address the numerous descriptions of surrounding objects and backgrounds within images, images are parsed into scene graphs, which are treated as fine-grained image representations to analyze the relations of these objects with a graph structure. On the other hand, following previous dependency-based RE methods (Guo et al., 2019), we parse the queries into syntax dependency trees, which can provide information for dual-modality graph alignment. As a result, we propose a graph learning-based method named DUMOGA to align

the semantics in queries and the information in images, aiming at efficiently locating the target instances. The whole structure of our proposed DUMOGA is shown in Fig 4.

## 3.1 DUAL-MODALITY GRAPH REPRESENTATION

**Semantic Scene Graph Generation for Vision.** We employ a classic SGG model (e.g. VCTree (Tang et al., 2019)) to parse images into scene graphs. The whole process can be formulated as:

$$P_r\left(\mathcal{G} \mid \mathcal{I}\right) = P_r\left(\mathcal{S}, \mathcal{O}, \mathcal{R} \mid \mathcal{I}\right). \tag{1}$$

An image $\mathcal{I}$ can be segmented into a set of masks $\mathcal{S}$. Each mask is associated with an object with class label $\mathcal{O}$. A set of relations $\mathcal{R}$ between objects are predicted. We construct the node set $N_i = \{O_j | j \in [1, n]\}$ of the scene graph, $O_i$ is the $i_{th}$ object detected in the image. $n$ represents the number of detected object, and we set the maximum number of detected object for each image to $N$. For the edge set $E_i = \{R_{a,b} = (I_a, I_b) | a \in [1, n], b \in [0, n-1]\}$, $R_{a,b}$ denotes the $a_{th}$ object related with the $b_{th}$ object. As a result, the scene graph can be formulated as follows:

$$G_i = (N_i, E_i). \tag{2}$$

**Syntax Dependency Graph for Text.** We use the syntax dependency tree to explore the dependency between words in the sentence. Following (Peters et al., 2018), we use ELMo to obtain the dependency tree for the input text after which each word from the text is connected by its governor and obtains its related dependency triple. For the node set $N_t = \{T_j | j \in [1, l]\}$ of the dependency tree, $T_j$ denotes the $j_{th}$ token in the sentence, and $l$ represents the total length of the sentence. For the edge set $E_t = \{G_{a,b} = (T_a^*, T_a) | a \in [1, l]\}$, $T_a^*$ denotes the governor of the $a_{th}$ token, and the graph representation for the sentence can be formulated as follows:

$$G_t = (N_t, E_t). \tag{3}$$

## 3.2 DUAL-MODALITY ALIGNMENT

To accurately locate the target instance using the query sentence, the gap between visual and semantics needs to be bridged. Thus, we propose the graph and feature alignment process to efficiently align visual and language domains.

**Graph Alignment.** We approximate the node embedding matrix $\widetilde{P}$ by factorizing a similarity matrix of the node identities, and align nodes between the above two graphs by greedily matching the most similar embeddings from the other graph.

Following (Zheng et al., 2021), we combine $N_i$ and $N_t$, and count both in and out degrees of k-top neighbors for each node. Then we compute the similarity between every two nodes, getting a $n \times n$ similarity matrix $\widetilde{P}$. After that we subsets $P_1$ and $P_2$ from $\widetilde{P}$. The $P_1$ and $P_2$ denote separate representations for nodes in $G_i$ and $G_t$.

$$\widetilde{P}_1, \widetilde{P}_2 = D\left(N\left(\widetilde{P}\right)\right), \tag{4}$$

where $D$ denotes the dividing operation of $P$ by the number of nodes in $N_i$ and $N_t$ in order, and $N$ is normalization operation. When finishing graph structure alignment, we calculate the similarity between node $i$ from $\widetilde{P}_1$ and node $j$ from $\widetilde{P}_2$ using the formula below:

$$a_{ij} = exp(-\left\|\widetilde{P}_1[i] - \widetilde{P}_2[j]\right\|_2^2). \tag{5}$$

With the similarities between nodes, the two graphs are transformed into a similarity map $\alpha$, which can be formulated as:

$$\alpha = (a_{ij})_{|V_1| \times |V_2|}, \tag{6}$$

where $a_{ij}$ represents the structural similarity between the $i_{th}$ word of the input text and the $j_{th}$ object of the input image.

**Feature Alignment.** Though graph structure is efficient, the information within is not enough for the task. We take the advantage of visual features from images and word embeddings from sentences to promise a fine-grained searching space. Specifically, for each image $I$, we get its visual feature $F_i$

Table 2: Comparison with state-of-the-art methods in terms of overall IoU on three benchmark datasets. *GA* represents Graph Alignment, *FA* represents Feature Alignment, and *Full* represents Graph Alignment + Feature Alignment.

| Method | Backbone Model | Refer Image Segmentation | | | | | |
|---|---|---|---|---|---|---|---|
| | | mIoU | P@0.3 | P@0.4 | P@0.5 | P@0.6 | P@0.7 |
| MAttNet (Yu et al., 2018) | ResNet-101 | 8.00 | 9.61 | 7.90 | 6.15 | 5.51 | 4.58 |
| VPD (Zhao et al., 2023) | U-Net | 24.0 | 29.5 | 27.5 | 23.8 | 21.5 | **19.3** |
| LAVT (Ding et al., 2021) | SWIN-B | 21.2 | 26.6 | 21.6 | 17.1 | 13.7 | 10.9 |
| UNINEXT (Yan et al., 2023) | ResNet-50 | 19.8 | 22.3 | 21.8 | 21.0 | 19.9 | 19.2 |
| DuMoGa (*GA*) | ResNet-50 | 15.0 | 19.5 | 18.3 | 16.7 | 14.4 | 11.7 |
| DuMoGa (*FA*) | ResNet-50 | 16.4 | 21.1 | 19.4 | 17.8 | 15.5 | 13.4 |
| DuMoGa (*FULL*) | ResNet-50 | **24.4** | **31.8** | **29.7** | **26.8** | **23.1** | **19.3** |

from backbone model (e.g. ResNet-50 (He et al., 2016)). For each sentence $Q$, we get its semantic embedding $F_l$ using BERT (Devlin et al., 2019). We treat $F_i$ as the query while $F_l$ is treated as the key and value, completing the attention process below:

$$R^a = \text{Softmax}\left(\frac{qk^T}{\sqrt{d}}\right)v, \tag{7}$$

where d denotes the dimension of $F_l$.

**Feature Fusion and Prediction.** After obtaining the results from graph alignment and feature alignment processes, we fuse them to make the final prediction.

$$R^f = MLP\left([R^a; \alpha F_l]\right), \quad s = j^* = \underset{1 \leq j \leq N}{\text{Argmax}}(R_1^f, ... R_j^f, ..., R_N^f), \tag{8}$$

Where $[;]$ denotes concatenation, and we use the MLPs to project the feature to $N$ dimension. $s$ denotes the output of the referring objects from the input with the largest probability.

## 3.3 Training Loss

For the $N$ detected objects in the image, we match their masks with the ground truth mask and obtain the $g_{th}$ object that share the most intersection over union with the ground truth. To maximize the probability on the $g_{th}$ dimension of $R^f$, we employ a cross-entropy loss function below:

$$L = \sum_{i=1}^{N}[-K^i log(R_i^f) - (1 - K^i)log(1 - R_i^f)]. \tag{9}$$

$K$ represents a one-hot vector with all zeros except for the $g_{th}$ dimension, and $K^i$ denotes the $i_{th}$ dimension of $K$. As a result, the probability on the $g_{th}$ dimension of $R^f$ is maximized.

## 4 Experiment

### 4.1 Experimental Settings

We employ a VCTree (Tang et al., 2019) with ResNet-50 (He et al., 2016) as its backbone for scene graph generation and visual feature extraction. The maximum detected object in one image is set to 10. Specifically, the feature dimension for each of the detected object is 1024. For the query sentence, we append the [CLS] and [SEP] tokens to the beginning and end of the sentence respectively. Then a pre-trained BERT is used to generate sentence embedding and the dimension is set to 768. For our DuMoGa model, it is trained with Adamw optimizer, where we set the base learning rate at 2e-5 and the batch size at 64.

### 4.2 Evaluation Metrics

For the Referring Image Segmentation task, we follow previous works (Liu et al., 2017; Margffoy-Tuay et al., 2018) and adopt Precision@$X$ and IoU to verify the effectiveness. The IoU calculates

| Raw Image | LAVT | DUMOGA (Ours) | Ground Truth |
| :---: | :---: | :---: | :---: |

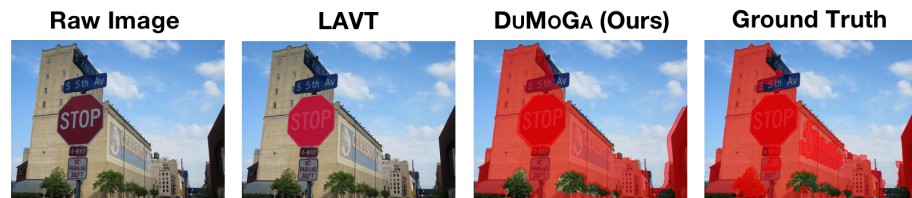

(a) Complex Query 1: In front of a stop sign and beside some trees, the building is standing under the sky.

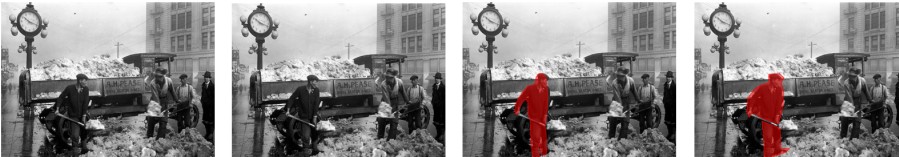

(b) Complex Query 2: The person is standing on the rock, in front of the truck, and beside the pavement.

Figure 5: Visualization result of our proposed DUMOGA model on RIS-CQ dataset. It is noteworthy that invalid prediction for LAVT was observed in the case of sample (b).

intersection regions over union regions of the predicted segmentation mask and the ground truth. The Precision@$X$ measures the percentage of test images with an IoU score higher than the threshold $X \in \{0.3, 0.4, 0.5, 0.6, 0.7\}$, which focuses on the location ability of the method.

### 4.3 PERFORMANCE COMPARISON

Table 2 shows the performance comparison with other SoTA methods on the RIS-CQ dataset, which includes the following models:

**LAVT** (Yang et al., 2022) is a Transformer-based framework for referring image segmentation. Unlike traditional approaches that fuse cross-modal information after feature extraction, LAVT incorporates language-aware visual encoding directly into the model architecture;

**VPD** (Zhao et al., 2023) proposed a new framework that exploits the semantic information of a pre-trained text-to-image diffusion model in visual perception tasks;

**UNINEXT** (Yan et al., 2023) reformulates diverse instance perception tasks into a unified object discovery and retrieval paradigm and can flexibly perceive different types of objects by simply changing the input prompts;

**MAttNet** (Yu et al., 2018) is a two-stage method that first extracts multiple instances by using an instance segmentation network Mask RCNN (He et al., 2017), then utilizes language features to select the target from the extracted instances. Due to the space limitation, we only list selected SoTA methods, more details can be found in **Supplementary Materials**.

As shown in Table 2, our method achieves remarkably superior performance on RIS-CQ dataset compared with other SoTA methods. Also our DUMOGA model is based on ResNet-50 (He et al., 2016), which is not so strong as Swin-B (Liu et al., 2021), DUMOGA achieves over 100% to 200% performance gain in all metrics compared with LAVT (Ding et al., 2021) on RIS-CQ dataset. With stronger backbone models, the performance of our DUMOGA model can be further improved.

We also provide the visualization of the output of our DUMOGA model in Figure 5. Compared with LAVT (Ding et al., 2021), our DUMOGA can accurately locate the referring object based on the comprehensive understanding of visual-textual data, rather than response to partial words in the query (such as the *stop sign* in Figure 5 (a)).

### 4.4 IN-DEPTH ANALYSIS

**Q1: How to control the annotation quality?** The annotation process for the RIS-CQ dataset spanned over a period of 6 months and involved the collaboration of 5 undergraduate students. To ensure the production of high-quality annotations, the annotators were supervised and adhered to specific principles. Firstly, all annotators underwent rigorous training before commencing the actual

annotation process. Secondly, separate annotators were assigned for query and object selection annotations. Annotators responsible for object selection were instructed to review the quality of queries, ensuring that unclear or poor queries were either rectified or eliminated. This approach simulated the evaluation process, guaranteeing the reasonableness of the queries while avoiding subjective annotations. Lastly, guidelines were provided for the maximal lengths of language queries, specifying limits of over 20 words for queries and over 5 for objects/background labels. We provide more details in the **Supplementary Materials**.

**Q2: What is the necessity of complex query? On the one hand**, current RIS models are sensitive to language queries. Even input two different queries which refer to the same object but contain information in different granularities, current RIS models will output different regions. So we need to construct a new RIS dataset with complex queries to validate the robustness of the proposed RIS models. **On the other hand**, it is not the final path to general RIS model if restricted to train models with downstream annotated benchmark datasets, such as RefCOCO with limited object classes and short query length. It's time to make a further step to real applications by combining with large pre-trained models and propose a novel and informative RIS dataset with more relations among objects.

## 5    WHAT TO DO NEXT?

With the emergence of large pre-trained models (such as GPT-4 (Bubeck et al., 2023) and SAM (Kirillov et al., 2023)), the semantic understanding ability of dealing with multi-modal data has been rapidly enhanced. Then, utilizing large pre-trained models is an irresistible trend due to their superiority in open-vocabulary scenarios. In the next step, we can develop lightweight RIS modules (based on the Graph Alignment and Feature Alignment modules in our DUMOGA model) to adapt large pre-trained models in a plug-and-play manner. Besides, on the basis of our proposed RIS-CQ dataset, we can explore more scenarios in real applications, such as referring object understanding in video and audio modality. And evaluate the robustness of proposed RIS models even with none or multiple referring objects according to the complex queries.

## 6    RELATED WORK

Referring image segmentation aims to segment a target region (*e.g.*, object or stuff) in an image by understanding a given natural linguistic expression, which was first introduced by (Hu et al., 2016). Early works (Liu et al., 2017; Li et al., 2018; Margffoy-Tuay et al., 2018) first extracted visual and linguistic features by CNN and LSTM, respectively, and directly concatenated two modalities to obtain final segmentation results by an FCN (Long et al., 2015). In MAttNet (Yu et al., 2018), Yu et al. proposed a two-stage method that first extracts instances using Mask R-CNN (He et al., 2017), and then adopts linguistic features to choose the target from those instances. Then, a series of works are proposed to adopt the attention mechanism. EFNet (Feng et al., 2021) designs a co-attention mechanism to use language to refine the multi-modal features progressively, which can promote the consistency of the cross-modal information representation. Some recent works leverage transformer (Vaswani et al., 2017) to deal with the RES task with satisfying performance. VLT (Ding et al., 2021) employs a transformer to build a network with an encoder-decoder attention mechanism for enhancing the global context information. All these methods are trained with RIS datasets with simple queries.

## 7    CONCLUSION

In this paper, we propose a novel benchmark dataset for Referring Image Segmentation with Complex Queries (RIS-CQ). The RIS-CQ dataset is of high quality and large scale, which challenges the existing RIS with complex, specific and informative language queries, and enables a more realistic scenario of RIS research. Besides, we propose a novel SoTA framework to task the RIS-CQ dataset, called dual-modality graph alignment model (DUMOGA), which effectively captures the intrinsic correspondence of the semantics of the two modalities. Experimental results and analyses demonstrate the necessity of our proposed dataset and the model.

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
