# Supplementary Material of Towards Complex-query Referring Image Segmentation: A Novel Benchmark

## 1 Performance Comparison

Table 1 shows the performance comparison with other SoTA methods on the RIS-CQ dataset, which includes the following models:

**LAVT** (Yang et al., 2022) is a Transformer-based framework for referring image segmentation. Unlike traditional approaches that fuse cross-modal information after feature extraction, LAVT incorporates language-aware visual encoding directly into the model architecture;

**VPD** (Zhao et al., 2023) proposed a new framework that exploits the semantic information of a pre-trained text-to-image diffusion model in visual perception tasks;

**UNINEXT** (Yan et al., 2023) reformulates diverse instance perception tasks into a unified object discovery and retrieval paradigm and can flexibly perceive different types of objects by simply changing the input prompts;

**MAttNet** (Yu et al., 2018) is a two-stage method that first extracts multiple instances by using an instance segmentation network Mask RCNN (He et al., 2017), then utilizes language features to select the target from the extracted instances.

As shown in Table 1, our method achieves remarkably superior performance on CIU-CQ dataset compared with other SoTA methods. Also our DuMoGa model is based on ResNet-50 He et al. (2016) and ResNet-101, which are not as strong as Swin-B Liu et al. (2021), DuMoGa achieves over 100% to 200% performance gain in all metrics compared with LAVT Ding et al. (2021) on RIS-CQ dataset. With stronger backbone models, the performance of our DuMoGa model can be further improved.

The DuMoGa framework is meticulously designed with a plug-and-play approach, aiming to address the process of efficient dual-modality alignment. The framework offers flexibility by allowing the

Table 1: Comparison with state-of-the-art methods in terms of overall IoU on three benchmark datasets. *GA* represents Graph Alignment, *FA* represents Feature Alignment, and *Full* represents Graph Alignment + Feature Alignment.

| Method | Backbone Model | Refer Image Understanding | | | | | |
|---|---|---|---|---|---|---|---|
| | | mIoU | P@0.3 | P@0.4 | P@0.5 | P@0.6 | P@0.7 |
| MAttNet | ResNet-101 | 8.00 | 9.61 | 7.90 | 6.15 | 5.51 | 4.58 |
| LAVT | SWIN-B | 21.2 | 26.6 | 21.6 | 17.1 | 13.7 | 10.9 |
| VPD | U-Net | 24.0 | 29.5 | 27.5 | 23.8 | 21.5 | 19.3 |
| UNINEXT | ResNet-50 | 19.8 | 22.3 | 21.8 | 21.0 | 19.9 | 19.2 |
| DuMoGa (*GA*) | ResNet-50 | 15.0 | 19.5 | 18.3 | 16.7 | 14.4 | 11.7 |
| DuMoGa (*FA*) | ResNet-50 | 16.4 | 21.1 | 19.4 | 17.8 | 15.5 | 13.4 |
| DuMoGa (*FULL*) | ResNet-50 | 24.4 | 31.8 | 29.7 | 26.8 | 23.1 | 19.3 |
| DuMoGa (*GA*) | ResNet-101 | 16.8 | 21.8 | 21.2 | 19.0 | 16.3 | 13.4 |
| DuMoGa (*FA*) | ResNet-101 | 17.5 | 21.6 | 20.3 | 18.7 | 16.9 | 14.7 |
| DuMoGa (*FULL*) | ResNet-101 | **25.3** | **32.4** | **30.4** | **28.0** | **25.2** | **20.5** |

substitution of the VCTree Tang et al. (2019) method, employed for scene graph generation, and the visual backbone, which is responsible for extracting visual feature representations, with alternative methodologies and backbone architectures. This adaptability makes the DuMoGa framework compatible with a wide range of approaches, thereby accommodating the evolving landscape of research in the field. By leveraging different backbone models, as demonstrated in Table 1, the DuMoGa framework exhibits its versatility. Through the utilization of more sophisticated feature extraction techniques, such as employing ResNet-101 instead of ResNet-50, the DuMoGa framework achieves superior performance across various evaluation metrics on the RIU dataset. This empirical evidence underscores the efficacy of incorporating more fine-grained feature extraction methods within the DuMoGa framework.

In essence, our DuMoGa framework exhibits excellent adaptability in an era characterized by the emergence of diverse feature extraction methods. For instance, the SAM-series of works Kirillov et al. (2023) have showcased remarkable prowess in instance segmentation tasks, yielding exceptional performance. Leveraging the advantages of the SAM models, they can serve as robust backbone models for the DuMoGa framework. This integration not only enhances the quality of generated scene graphs but also improves the extraction of visual features, leading to more accurate and comprehensive inference results. The integration of SAM into the DuMoGa framework represents a significant advancement in the field, showcasing the potential for synergistic collaboration between different methods. It demonstrates how the adaptability and extensibility of the DuMoGa framework facilitate the integration of state-of-the-art techniques, ultimately leading to improved performance and promising avenues for further research in the community.

**Effectiveness of each component in our DUMOGA method.** As shown in Table 1, we list three variants of our DUMOGA model. DUMOGA (*GA*), DUMOGA (*FA*), and DUMOGA (*Full*) represent DUMOGA with only graph alignment, DUMOGA with only feature alignment, and DUMOGA with graph alignment + feature alignment, respectively. To evaluate the effectiveness of graph alignment process, we train a DUMOGA (*GA*) only considers the information contained in scene graphs and semantic tree graphs, which only have a few MLPs to complete the task. The same evaluation process for our feature alignment process, we only use the visual feature extracted from the SGG model and the sentence embedding from BERT. These two simple models still outperform traditional RIS methods by a great margin. The reasons is that as the complexity of query sentence increasing, traditional methods may be mislead by the sophisticated logic and finally get low performances, however, the parsed dependency tree graph is likely to shares higher similarity with the generated scene graph for its richer information within. As a result, our method easily achieves better performance.

## 2   REFERRING QUERY GENERATION

Figure 1 depicts the procedural flow of generating referring queries using the outputs of a scene graph model. **Initially**, we engage in a dialogue with ChatGPT using Prompt A as the opening prompt. Prompt A serves the purpose of clarifying the task we seek ChatGPT's assistance with and outlining the associated requirements. Additionally, examples are provided to harness the powerful in-context learning capabilities of ChatGPT, aiming to achieve reasoned inferences that align with our specified needs. **Subsequently**, we instruct ChatGPT, following Prompt A, regarding the object we intend to describe and its relevant relations to other objects. This enables ChatGPT to generate descriptions of the designated objects as per the instructions outlined in the prompt. It is noteworthy that, to alleviate the inferential burden on ChatGPT, the triplets generated by the scene graph model (e.g., <person, book, holding>) are directly transformed into complete sentences (e.g., "the person is holding the book"). Moreover, to differentiate between objects of the same category, numerical suffixes such as "person2" or "cow3" are assigned. This facilitates relatively precise object descriptions leveraging relations, thereby significantly reducing the need for labor-intensive manual annotation of specific object attributes. **Finally**, we perform a rewrite on the initial answer generated by ChatGPT in Answer B, removing the numeric suffix corresponding to the object (e.g., "<person1 is next to person2" becomes "the person is next to another person," and "<the floor has person1, person2, and person3 standing on it" becomes "the floor has multiple people standing on it").

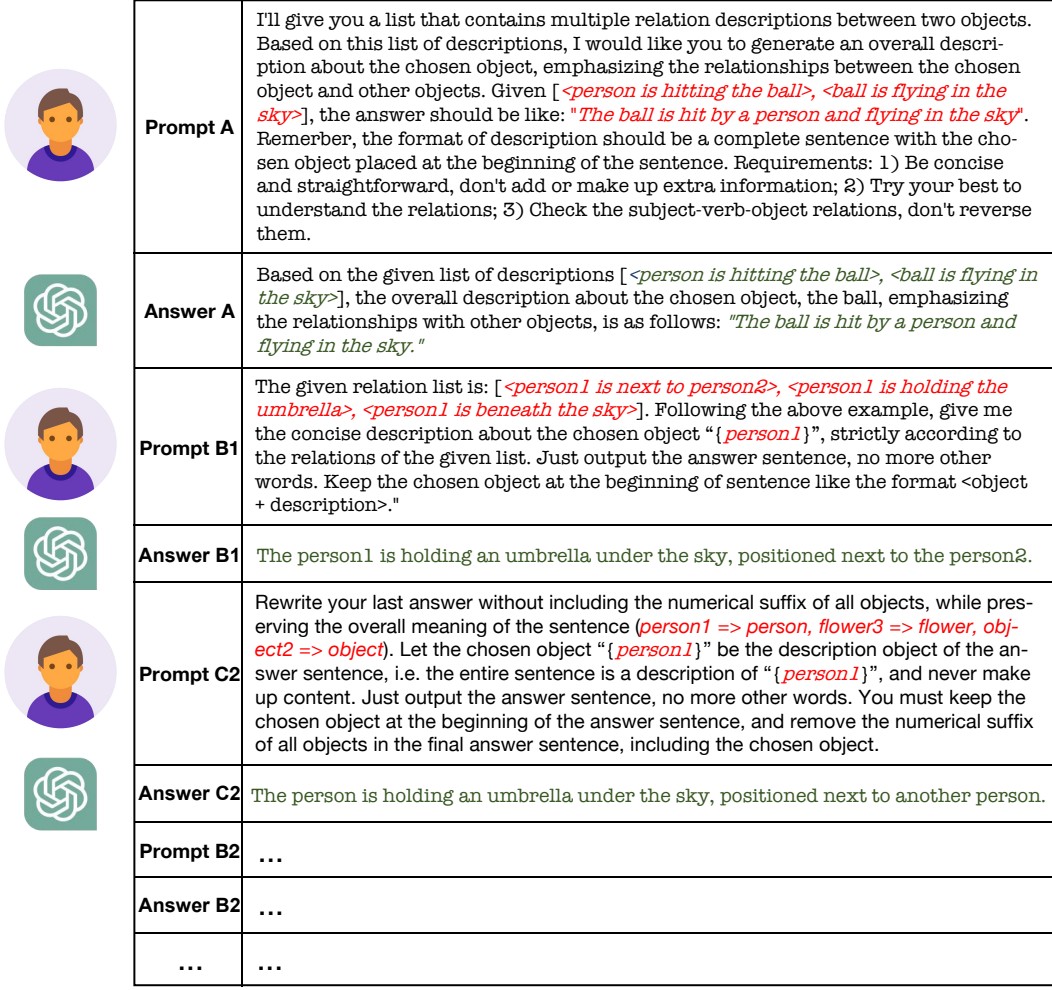

Figure 1: An example of generating descriptions for a specified object based on the relations between objects provided by a scene graph model using ChatGPT.

## 3 POTENTIAL IMPACT AND LIMITATIONS

**Potential Impact.** Referring Image Segmentation can track people in the scene and understand their relations with the surrounding objects via language description, it has the potential to be used to violate people's privacy. The possible application to the surveillance system can also bring negative societal impacts.

**Limitation.** In this paper, we explore referring image segmentation with complex queries for only one object. However, referring multiple objects with complex language queries is also an important task in real applications, which we will leave as future work.