# OpenReview forum: "Towards Complex-query Referring Image Segmentation: A Novel Benchmark"
_ICLR.cc/2024/Conference — ICLR 2024 Conference Withdrawn Submission_

### Official Review · Reviewer_iqmc · 2023-10-30

**Soundness:** 4 excellent
**Presentation:** 3 good
**Contribution:** 4 excellent
**Rating:** 8
**Confidence:** 5

**Summary:**

This research introduces the "Complex-query Referring Image Understanding (RIS-CQ)" benchmark, aiming to elevate referring image segmentation associated with intricate language queries. A groundbreaking dual-modality graph alignment model named "DuMoGA" is unveiled. This model adeptly aligns semantic graphs between textual and visual elements. Remarkably, DuMoGA surpasses current methodologies on RIS-CQ, showcasing its proficiency in deciphering multifaceted language queries.

**Strengths:**

1. The paper addresses a pivotal challenge in referring image segmentation, specifically targeting intricate linguistic queries. By launching the RIS-CQ benchmark, the authors compellingly substantiate the essence of their exploration. This benchmark holds the potential to be an instrumental anchor for subsequent investigations in this realm.

2. A key contribution in the paper is the introduction of the RIS-CQ dataset. This is a commendable gift to the academic domain. The amalgamation of this fresh dataset, paired with a potent model for referring image segmentation, augments the arsenal of tools and resources at the disposal of researchers.

3. The proposed DuMoGA model presents an innovative strategy. The design is quite interesting. It excels in performance against existing paradigms. Its remarkable prowess in navigating complex linguistic contexts underscores its capacity to amplify image comprehension in intricate scenarios.


Overall, I like this work. The paper presents a strong and significant contribution to the field of referring image segmentation. Addressing a challenging problem, introducing a new benchmark, and proposing an effective model for complex language queries, the paper holds considerable promise for the research community. With some minor enhancements, this work would make a valuable addition to the community.

**Weaknesses:**

The paper might benefit from further improvements by addressing some of the minor issues I found:

1. While the paper presents a strong case for the effectiveness of the proposed method, it would be beneficial to include some negative or challenging cases in the experimental section. Providing examples where the approach may have limitations or difficulties would add depth to the evaluation.

2. The related work section could be improved by incorporating more recent works in the field. This would provide readers with a more comprehensive overview of the current research landscape, demonstrating the paper's position in the context of recent advancements.

3. In terms of language expression, further polishing might be needed. Some phrasings require enhancement.

**Questions:**

1. Could the authors provide further details on the methods used to control and measure the quality of annotations for the RIS-CQ dataset? A discussion of the annotation process, guidelines, and inter-annotator agreement would enhance the understanding of dataset quality.

2. It would be helpful to elaborate on the specific advantages offered by the proposed method compared to other SOTA methods. Highlighting the unique strengths and capabilities of DuMoGA relative to existing models would further emphasize its contributions.

---

### Official Review · Reviewer_V6x1 · 2023-11-04

**Soundness:** 2 fair
**Presentation:** 3 good
**Contribution:** 2 fair
**Rating:** 3
**Confidence:** 5

**Summary:**

The paper proposes a new dataset for referrer image segmentation and a baseline on this dataset. The new dataset, namely RIS-CQ, built upon RefCOCO and Visual Genome contains complex text queries for each image. The complex text queries contain more specific information, generated from GPT3.5 and manually refined. Since other works performed poorly on the new dataset, they propose a dual-modality graph alignment model. The model makes use of two types of information: graph alignment of sense graph and text sequence, and feature alignment of image embedding and text embedding. The baseline mode outperforms previous work on the new dataset.

**Strengths:**

- They provide a new dataset with complex text queries. The complex text queries are generated based on GPT 3.5 with inputs being the triplets of object relationships.
- The baseline is intuitive and works well. They fuse multimodal information Graph alignment: scene graph and text sentence.

**Weaknesses:**

1. Only a simple comparison with RefCOCOg is given. What are the advantages compared with RefCOCOg? Is there essential difference between query lengths of 8.43 and 13.18?
2. The comparison with `PhraseCut' is missed, which is also a large-scale dataset with complex queries.
3. The key in the proposed method is scene-graph-based cross-modal alignment. However, this way is widely used in cross-modal retrieval works.
4. How to get mask annotations?
5. VCTree is a method to generate bounding boxes. How to generate mask predictions?

**Questions:**

Please see Weaknesses.

**Details Of Ethics Concerns:**

(Minor) Since the text descriptions are generated from GPT3.5, it would be great to discuss the ethical problem that may be caused by generative AI.

---

### Official Review · Reviewer_8JWc · 2023-11-08

**Soundness:** 1 poor
**Presentation:** 2 fair
**Contribution:** 2 fair
**Rating:** 3
**Confidence:** 5

**Summary:**

This work proposed a new benchmark for the referring expression task, a system (DuMoGa) that models the task using the two modality semantic graphs and attempts to align them. Moreover, in-depth analyses are shown based on the dataset and system that provides findings.

**Strengths:**

- It is not just the dataset a good contribution to the referring expression community. But also, the way that the dataset is constructed by using ChatGPT to enrich the expression and reduce the human workforce necessary to annotate this complex task is clearly a new trend and a good thing to use.
- The dataset is bigger than the previous dataset in the quantity of images and the length of the queries.

**Weaknesses:**

- Not sure about the system statement Sec 3: "Unlike classic dense predictions models with complex design in RIS, which take expensive computational power to make inference, the graph learning-based architecture provides efficient training process and promising results". Graph learning-based architecture needs a couple of other methods that are complex to get the scene graph and the dependency trees. The computational power to make inferences per image requires three methods to run over the image (DP, SGG and the DuMoGa). I consider this a bold statement with no supporting experiments in the paper regarding inference or training time and computational requirements.
- The in-depth analysis needs more experiments that could support the statements and bring dome insights.

**Questions:**

- I reckon ChatGPT4v can obtain more contextual information from the images, which could help to solve those ambiguous references. Chapter 2, step-3.
- Although queries are semantically richer than other datasets, images come from combining two previously existing datasets (VG and COCO). Could it be that the lengths of queries in previous datasets are short because the complexity of the images is less? I wonder if reannotating those datasets is enough to create a good benchmark for RIS. It is a step forward, but it would be good to know how complex are the images in the previously mentioned datasets. Maybe report the average amount of detected objects in the images, similar to the statement "85.3% text queries have short forms (i.e., with length < 5 words)" but on the image side.
- "all annotators underwent rigorous training before commencing the actual annotation process." what is the rigorous training?
- " Even input two different queries which refer to the same object but contain information in different granularities, current RIS models will output different regions." no qualitative or quantitative experiments show this issue or, at least, a reference. In this regard, an analysis of semantically similar queries using METEOR or SPICE and then show how many of the queries that are similar by certain threshold ground different objects would support that statement.

---

### Official Review · Reviewer_ruSq · 2023-11-10

**Soundness:** 1 poor
**Presentation:** 1 poor
**Contribution:** 2 fair
**Rating:** 3
**Confidence:** 2

**Summary:**

The paper leverages the improvement in the semantic understanding capability of large pre-trained models to propose a RIS benchmark
that resembles real-world applications -> RIS-CQ, which challenges the existing RIS with complex queries, and enables a more
realistic scenario of RIS research.
The authors also provide a baseline on this dataset.

**Strengths:**

The paper proposes a method of annotation generation that is scaleable by leveraging existing foundational methods.
The proposed benchmark dataset has more complex queries than existing benchmarks

**Weaknesses:**

The results on the proposed DUMOGA method are not clear from table2. It's not clear exactly what datasets the model was evaluated on
Ideally, it should be evaluated on the proposed benchmark as well as on existing benchmarks to help calibrate the improvements

**Questions:**

Can table 2 be clarified further? The table caption says 'on three benchmark datasets', but it's not clear which ones and where